# Supervised Fine-Tuning for Unsupervised KPI Anomaly Detection for Mobile Web Systems

## ABSTRACT

With the rapid development of cellular networks, wireless base stations (WBSes) have become crucial infrastructure for mobile web systems. To ensure service quality, operators constantly monitor the operation status of WBSes and deploy anomaly detection methods to identify anomalies promptly. After the deployment of anomaly detection methods, operators periodically collect feedback, which holds significant value in improving anomaly detection performance. In real-world industrial environments, the frequency of false negative feedback is usually very low, and the newly generated data's distribution can differ significantly from that of the original training data. Therefore, the feedback-based performance improvement of the previously proposed methods is limited. In this paper, we propose *AnoTuner*, which incorporates a false negative augmentation mechanism to generate similar false negative feedback cases, effectively compensating for the low feedback frequency. Additionally, we introduce a Two-Stage Active Learning (TSAL) mechanism that minimizes data contamination issues caused by the difference between the distribution of feedback data and that of the training data. Experiments conducted on the real-world data collected from a top-tier global Internet Service Provider (ISP) demonstrate that the performance improvement of *AnoTuner* after feedback-based fine-tuning is significantly higher than that of the best baseline method. Our codes are released anonymously at https://anonymous.4open.science/r/AnoTuner/.

**Relevance Statement**: Our paper is highly relevant to the track "Internet systems, applications, and Web of Things (WoT) applications", focusing on maintaining the mobile web system reliability. Instead of merely using a Web artifact, our work addresses a genuine Web-centric challenge: optimizing anomaly detection in WBSes. By introducing *AnoTuner*, we offer a unique solution to real-world data challenges faced by ISP. This study addresses a core concern in today's Internet infrastructure.

## CCS CONCEPTS

• **Computing methodologies** → **Anomaly detection**; • **Networks** → *Network management*; • **Information systems** → *Data mining*.

## KEYWORDS

Anomaly detection, Multivariate Time-Series, System Reliability

## 1 INTRODUCTION

A Wireless Base Station (WBS), also known as a cell tower or cellular tower, is a fundamental component of mobile web systems. It serves as a central hub for wireless communication within a specific geographic area. With the rapid development of mobile networks, WBSes play a critical role in ensuring reliable and efficient communication World-Wide Web [5, 7, 12, 13, 15, 18]. However,

the occurrence of anomalies or malfunctions in these WBSes can significantly impact network performance, leading to degraded user experience, service disruptions, and even massive economic losses [3].

To ensure service quality, operators from Internet Service Providers (ISPs) constantly monitor the operational status of WBSes and deploy anomaly detection methods to identify anomalies promptly [3]. WBSes generate a substantial amount of monitoring data on a daily basis. These data include Key Performance Indicators (KPIs) such as wireless connection rate, interference level, Radio Resource Control (RRC) connection requests, E-UTRAN Radio Access Bearer (E-RAB) establishment success rate, Control Channel Element (CCE) utilization rate, etc. They are in the form of multivariate time series (MTS) reflecting the operational status of a WBS during a particular period [3]. Quick and accurate detection of MTS anomalies contributes to the prompt identification of issues, preventing more severe failures from occurring and enabling downstream methods to swiftly pinpoint the causes of malfunctions, facilitating failure mitigations as soon as possible [3, 10, 27].

After the deployment of an anomaly detection method in practice, it is common to collect feedback about the method. These feedback data include false positives (false alarms) and false negatives (missed alarms). The normal operation of WBSes is exceptionally vital for ISPs. Therefore, operators often calibrate the model to be sensitive to anomalies to reduce false negatives. This leads to a paradox: in real-world industrial environments, operators are more concerned with the information carried by false negatives, yet the feedback data collected contains few instances of false negatives.

Feedback data serve as crucial sources of information for improving the anomaly detection methods' performance. However, current MTS anomaly detection methods struggle to utilize feedback, especially the feedback of false negatives, effectively. Specifically, due to the enormous scale of WBSes, a massive amount of data is generated daily, making labeling data extremely expensive. Therefore, the mainstream MTS anomaly detection methods, such as Donut [27], LSTM-NDT [4], OmniAnomaly [23], AnomalyTrans [28], and Interfusion [11], are unsupervised. However, these methods waste information carried by the labels in operators' feedback. Semi-supervised learning, which applies labeled and unlabeled data to train an MTS anomaly detection model, seems to be a promising direction to utilize operators' feedback. Nevertheless, it fails to yield satisfactory performance improvement when utilizing the feedback data because of the following two challenges:

(1) **Scarce data**. Anomalies in WBS are infrequent, and due to operators' preference for anomaly detection methods' configurations, false negative feedback data are even rarer. It is difficult for semi-supervised MTS anomaly detection methods to learn effectively from the scarce data of false negative feedback, which operators are seriously concerned.

(2) **Biased data distribution**. The distribution of the feedback data collected after deploying an MTS anomaly detection method can differ significantly from that of the training data because of software/hardware upgrades and configuration changes [14, 19, 29] (see Section 2.2 for more details). This discrepancy may cause model contamination during the feedback-based performance improvement.

To address the above challenges, we propose *AnoTuner*, a Supervised **Ano**maly **Tuner** for Unsupervised KPI Anomaly Detection. *AnoTuner* is trained in an unsupervised manner and can be fine-tuned using the feedback data with our novel **Label-Aware Evidence Lower BOund (LAELBO) loss function**. Powered by the Conditional Variational Auto-Encoder (CVAE) [22] framework, *AnoTuner* possesses powerful data generation capabilities. Specifically, we design a **false negative augmentation** mechanism, generating more similar feedback based on existing false negative feedback to alleviate the scarcity of such feedback data. Moreover, we design a **Two-Stage Active Learning (TSAL)** mechanism that samples a small amount from historical data to mitigate the discrepancy between feedback data distribution and training data distribution, effectively avoiding data contamination. Experiments using real-world data collected from a top-tier global ISP show that the performance improvement of *AnoTuner* after feedback-based fine-tuning is significantly higher than that of the best baseline semi-supervised method, and this is achieved using the feedback data that constitutes only 0.74% of the test set. Due to double-blind reviewing, our codes are released anonymously at https://anonymous.4open.science/r/AnoTuner/.

Our contributions can be summarized as follows:

- We conducted a systematic study on the impact of feedback on improving the performance of MTS anomaly detection methods for WBSes. Based on that, we propose a novel Label-Aware ELBO (LAELBO) loss function to utilize the feedback data effectively.
- We propose a false negative augmentation mechanism to address the problem of the small data volume of false negative feedback when fine-tuning the MTS anomaly detection model, addressing the first challenge.
- We design a TSAL mechanism that effectively addresses the performance degradation issue caused by the biased data distribution, addressing the second challenge.
- Extensive evaluation experiments have been conducted using the real-world data of a large-scale ISP to verify *AnoTuner*'s performance.

## 2 PRELIMINERY

### 2.1 Background

In order to ensure the service quality of WBSes, operators monitor and collect various KPIs for each base station on a regular basis. In a WBS, if the pattern of one or several KPIs deviates from its historical norm or if there is a distortion in the physical relationship between the KPIs, it is typically considered that an anomaly has occurred. The causes of these anomalies can be diverse, such as an incorrect software update, physical damage to the base station itself, or malicious attacks on the base station. After interviews with ISP operators, we have identified 25 widely used KPIs for WBSes listed in Appendix A.

Such data can be formalized as multivariate time series data shown in Figure 1, which can reflect the operational status of each WBS. Operators typically deploy anomaly detection methods for the WBSes. Traditional anomaly detection methods are usually based on rules. With the advancement of machine learning techniques and their demonstrated powerful performance across various domains, machine learning methods have become the primary approach for WBS anomaly detection in the industry [3, 6, 12, 15, 17].

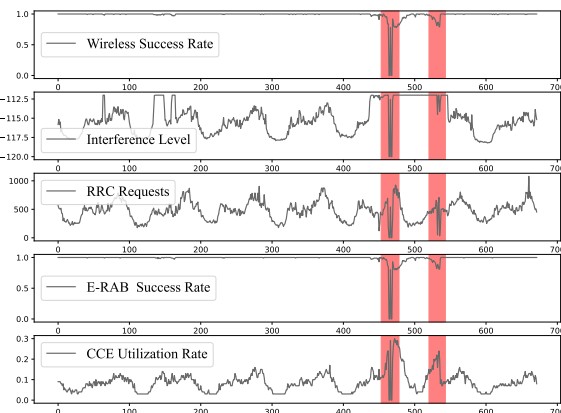

**Figure 1: An example of a WBS's multivariate time series. The red rectangles mark the anomaly interval.**

However, in a real-world industrial environment, completing a method's training process does not signify the end of the method's deployment. Deployed methods will inevitably encounter issues in production, generating false positives (false alarms) and false negatives (missed alarms). When utilizing rules for anomaly detection, operators often patch existing rules based on these feedback data. However, such an approach is infeasible for methods based on machine learning. For machine learning methods, the standard practice in the industry is to collect sufficient feedbacks over a relatively short period, say a week, to fine-tune the model and to retrain the model over a more extended period, such as a quarter [10]. Given that model training costs significantly more than fine-tuning, ISPs tend not to retrain anomaly detection models very often. Therefore, if fine-tuning using feedback fails to resolve the issue effectively, the problem of the model will persist until the next retraining session. During this period, similar false positives or negatives may reoccur, undoubtedly causing a great deal of unnecessary trouble for operators and reducing the credibility of the anomaly detection results.

### 2.2 Motivation

Anomaly detection plays a crucial role in the reliable operation of WBSes. Accurate anomaly detection can help operators promptly identify potential issues in the WBSes and mitigate them as soon as possible, preventing minor anomalies from escalating into severe failures. Moreover, the results of anomaly detection can be forwarded to downstream methods for subsequent analysis, such as feeding into root cause localization methods to automatically pinpoint the causes of WBS failures.

Fine-tuning anomaly detection methods using feedback data has been a practice since the era of rule-based anomaly detection. Patches for rules can effectively resolve similar issues based on the feedback data. Moreover, in the era of machine learning, fine-tuning with feedback data is also a common behavior. However, in WBS anomaly detection, due to the low frequency of anomalies and biased feedback data distribution [10], existing methods struggle to effectively utilize online feedback in WBSes.

There are three main reasons why we focus on the fine-tuning of anomaly detection models based on feedback, especially feedback of false negatives. Firstly, feedback derived from real-world applications of an MTS anomaly detection method significantly contributes to the enhancement of the method. Fine-tuning the method through feedback is an excellent way to incorporate the domain knowledge of operators into the method. Figure 2 shows a real-world experiment that we conducted. We collected a week's worth of feedback on our deployed anomaly detection method (*i.e. AnoTuner*) from a certain region of a top-tier global ISP and fine-tuned the method accordingly. After this feedback-based fine-tuning, both the number of false positive and false negative points of the model significantly decreased. Notably, these feedback data used for fine-tuning only constituted 0.6% of the total dataset.

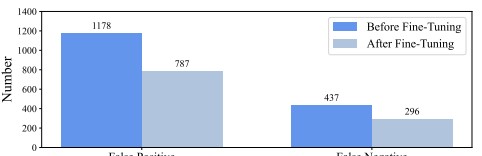

**Figure 2: Comparison of the numbers of false positives and false negatives before and after feedback-based fine-tuning.**

Secondly, the reliable operation of WBSes is crucial for ISPs, hence, the impact of false negatives on WBSes is particularly severe. For instance, if an anomaly in a base station is not timely identified due to a false negative, it could lead to delayed mitigations, causing the anomaly to escalate into a severe failure. Therefore, it is worthwhile to pay extra attention to false negative feedback in anomaly detection for WBSes.

Thirdly, the proportion of false negative and false positive feedback on the MTS anomaly detection methods is imbalanced. The quantity of false positive feedback is typically 2 to 10 times that of false negative feedback. The imbalance between false negatives and false positives exacerbates the issue of feedback scarcity, making the scarcity of false negatives even more pronounced. Given that a machine learning-based anomaly detection method generally learns patterns from large amounts of data, this imbalance has resulted in the underutilization of false negative feedback, thereby wasting the valuable information that could be used to improve the method's performance.

## 3 METHODOLOGY

In this section, we introduce our innovative approach to address the challenges encountered in Section 1. We first provide an overview of our method, followed by a detailed description of our design improvements in the anomaly detection model and the LAELBO

loss function to better utilize feedback data. Subsequently, we analyze the false negative feedback in the WBS and present our False Negative Augmentation technique to mitigate the scarce data problem. Finally, we introduce the Two-Stage Active Learning (TSAL) utilized to eliminate the Biased Distribution issue.

### 3.1 Model overview

The overall workflow of *AnoTuner* is as depicted in Figure 3. The workflow of *AnoTuner* is divided into three phases: unsupervised training and deployment, feedback collection, and supervised fine-tuning.

During the feedback collection process, operators periodically provide feedback on false positives or false negatives reported by the anomaly detection method. In practical environments, this cycle is typically one week.

In the feedback-based fine-tuning phase, *AnoTuner* differs significantly from current anomaly detection methods. Firstly, we do not use the feedback data directly for fine-tuning but go through two crucial steps: false negative augmentation, and TSAL. We describe these two steps in detail below. It is worth noting that these two steps are related to the design of our base model and cannot be directly applied to other anomaly detection methods. Nevertheless, they can still provide a reference value for the design of other anomaly detection methods.

Next, we will provide a detailed description of our base model design, false negative augmentation, and TSAL.

### 3.2 Label-Aware ELBO Loss Function

Our model adopts the framework of the Conditional Variational Auto-Encoder (CVAE) [22], trained via an widely used encode-decode method. The fine-tuning process in *AnoTuner* differs from the training process. The training process of *AnoTuner* is unsupervised, while the fine-tuning process is supervised. This is because if the same unsupervised mode used during training is still applied in the fine-tuning process, the model cannot effectively learn from the false negative data, as the false negative data itself represents anomaly patterns that the model mistakenly learned from the training set. To achieve supervised fine-tuning, we have made improvements to the loss function of the CVAE loss function called Label-Aware Evidence Lower BOund (ELBO), as shown in (2).

The original ELBO loss function of the CVAE typically comprises two components: the reconstruction loss and the KL divergence.

Given a data point $x$, its associated condition $c$, and its latent variable $z$, the objective of the CVAE is to maximize the ELBO given by:

$$\text{ELBO} = \mathbb{E}_{q_\phi(z|x,c)}[\log p_\theta(x|z,c)] - \text{KL}(q_\phi(z|x,c)||p_\theta(z|c)) \quad (1)$$

$$= \mathbb{E}_{q_\phi(z|x,c)}[\log p_\theta(x|z,c)] - \int q_\phi(z|x,c) \log \frac{q_\phi(z|x,c)}{p_\theta(z|c)} dz$$

Where:

- $q_\phi(z|x,c)$ is the posterior probability distribution defined by the encoder, with parameters represented by $\phi$.
- $p_\theta(x|z,c)$ is the generative model defined by the decoder, with parameters represented by $\theta$.

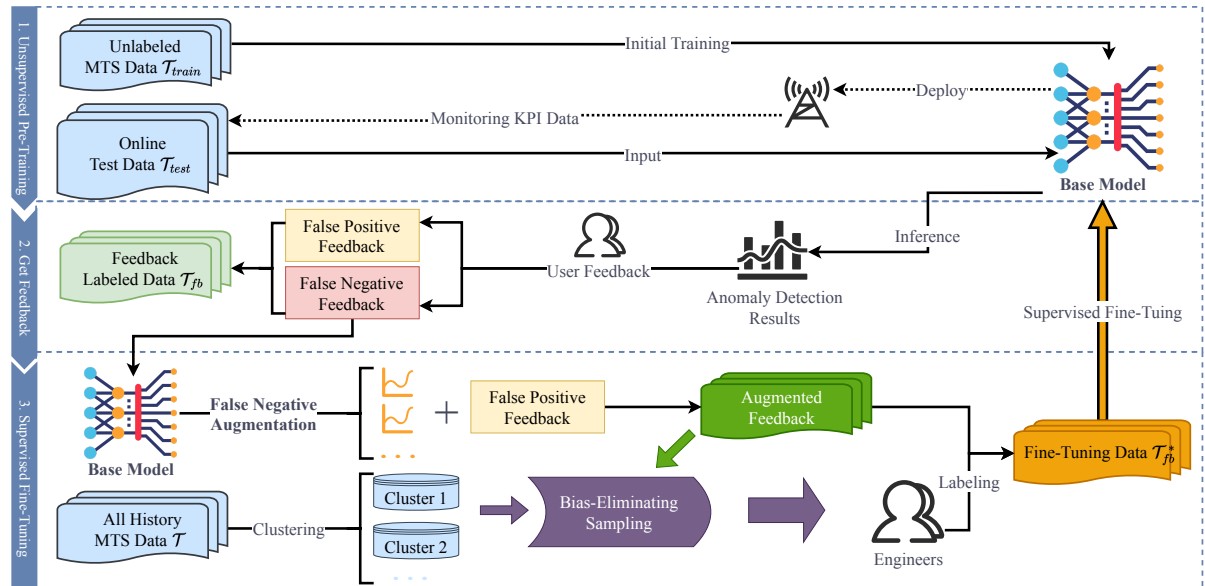

Figure 3: The overall workflow of *AnoTuner*.

- $p_\theta(z|c)$ is the prior probability distribution of the latent variable $z$, usually chosen to be a standard normal distribution.
- $KL(q_\phi(z|x,c)||p(z))$ is the KL divergence, measuring the difference between two probability distributions.

In practice, we typically do not compute the ELBO directly but minimize its negative value, which is equivalent to maximizing the ELBO.

In the proposed model, the given condition serves as an intrinsic feature of the data and does not encapsulate any labeling information. Consequently, the standard ELBO is not applicable to our framework. The fundamental principle of our approach is the categorization based on labels $y$. For instances where $y = 0$, our objective is to minimize the reconstruction error, thereby approximating the distributions $q_\phi(z|x,c)$ and $p_\theta(x|c)$ closely. Conversely, for $y = 1$, our goal is to amplify the reconstruction error, ensuring that the distributions $q_\phi(z|x,c)$ and $p_\theta(x|c)$ diverge significantly.

$$\mathcal{L}(x,c) = -\mathbb{E}_{z\sim q_\phi(z|x,c)}\left[\log p_\theta(x|z,c,y=0)\right]$$
$$+ \mathbb{E}_{z\sim q_\phi(z|x,c)}\left[\log p_\theta(x|z,c,y=1)\right] \quad (2)$$
$$- \int (2y-1)q_\phi(z|x,c)\log\frac{q_\phi(z|x,c)}{p_\theta(z|c)}dz$$

It is noteworthy that the label-aware ELBO is only used in feedback-based fine-tuning.

Subsequent experiments have shown that in the context of anomaly detection for WBSes, *AnoTuner* not only exhibits state-of-the-art performance but also can effectively utilize feedback for fine-tuning.

## 3.3 False Negative Augmentation

In the context of anomaly detection in WBSes, operators are reluctant to have the method miss any anomalies. As a result, the method is adjusted to be very sensitive to anomalies, leading to a relatively large number of false alarms. This results in a much smaller proportion of false negative data compared to false positives in the feedback. However, experimental results show that false negative feedback is crucial for fine-tuning. To better utilize false negative data for fine-tuning, an intuitive approach is to generate more similar cases using the existing feedback.

The idea of using existing data to generate more similar data is commonly referred to as data augmentation in the field of machine learning. In computer vision, data augmentation has been widely applied because images have relatively easy-to-understand physical meanings [2, 21, 26, 32]. One can use various digital image processing techniques to manipulate the images, generating more data while preserving their physical meanings. However, the data generated by WBSes is MTS data. Traditional data augmentation methods struggle to generate new data while ensuring the correctness of temporal and inter-KPI dependencies, which are characteristics of MTS data. This leads to traditional data augmentation introducing a large amount of noise into MTS data, unable to generate high-quality data.

To address this problem, we design a deep learning-based conditional generation for MTS. Thanks to our CVAE-based architecture, our *AnoTuner* can not only perform anomaly detection but is also inherently capable of generating time series data. Therefore, we do not need to train a separate generation model to produce more data; we can directly use the trained *AnoTuner* for generation. It is also worth noting that the primary cause of false negatives is the incorporation of anomalous data or noise into the training data. The model incorrectly learns the pattern of the anomalies during the unsupervised training stage, resulting in the reconstructed data being indistinguishable from the original data. This fact precisely illustrates that *AnoTuner* can effectively generate data similar to the

false negative data without worrying about a significant difference between the generated and real data.

Our process for false negative augmentation begins with encoding the false negative feedback data using the *AnoTuner* encoder, which results in the mean, $\mu$, and standard deviation, $\sigma$, of the distribution. Subsequently, we generate a set of latent variables, denoted as $\{z_1, z_2, ..., z_n\}$, by performing multiple samplings from a standard Gaussian distribution $\mathcal{N}(0, 1)$. The original false negative feedback data is then decomposed using STL, yielding seasonal and trend components. These components are combined to derive a conditional variable, $c$. In the final step, each latent variable $z_i$ is concatenated with the conditional variable and dispatched to the *AnoTuner* decoder for decoding, thereby producing the generated data. Concerning the volume of data to generate for false negative, our subsequent experiments suggest that the model accomplishes more effective learning when the ratio of false negative to false negative feedback data is approximately 1, see more details in Section 4.4.

Our experiments indicate that this false negative augmentation for false negative can not only generate high-quality feedback data but can also enhance the fine-tuning performance of the anomaly detection model.

## 3.4 Two-Stage Active Learning (TSAL)

A significant challenge encountered with feedback-based fine-tuning is the bias in feedback data. Our experimental analysis has identified two primary sources of this biased data distribution in feedback.

The first source of bias is the imbalance between false negative and false positive data. The second source of bias arises from the fact that the feedback data we use comprises data points that the current algorithm fails to process effectively because of software/hardware upgrades and configuration changes [14, 19, 29]. The biased data distribution can contaminate the model during fine-tuning leading to performance decay.

We have solved this first issue by false negative augmentation in Section 3.3. To address the second issue, we propose a Two-Stage Active Learning (TSAL). The core idea of this mechanism is to balance the discrepancy in feedback data distribution by sampling a small amount of historical data, making the data more in line with the independent and identically distributed assumption in machine learning. This mechanism is divided into two stages: the first is the cluster-based filter, and the second is bias-eliminating sampling.

The primary goal of the cluster-based filter in the first stage is to identify data from the historical data that differs from the existing feedback data. The first challenge we need to tackle in this stage is the subsequence clustering of MTS data. For this purpose, we are inspired by OmniCluster [30], a SOTA approach to MTS clustering. Instead of separately training an OmniCluster, we use the encoder from *AnoTuner* to encode both historical and current feedback data.

There are two reasons for this decision. Firstly, the core objective of training OmniCluster is to obtain an encoder, and the encoder of *AnoTuner*, after its training phase, already exhibits robust feature extraction capabilities, enabling it to adequately encode time series. Secondly, the goal of our clustering is to identify data that *AnoTuner* perceives as different from the existing feedback data. Therefore, using *AnoTuner*'s encoder for encoding is more likely to accomplish

this goal. After encoding, we apply the OmniCluster structure to cluster all the historical data. We then attempt to assign feedback data to these existing clusters, and selected clusters that have not been assigned for the next stage.

---

**Algorithm 1** Bias-Eliminating Sampling

**Input:** Clusters $C = \{C_1, C_2, ..., C_n\}$, Feedback data $\mathcal{T}_{fb}$;
**Output:** $\mathcal{T}_{fb}^*$ dataset used for fine-tuning;
1: assign each data point in $\mathcal{T}_{fb}$ to $C$;
2: get unassigned clusters $C^0 = \{C_1^0, C_2^0, ..., C_k^0\}$ and assigned clusters $C^1 = \{C_1^1, C_2^1, ..., C_l^1\}$;
3: **if** $C^1 \neq \emptyset$ **then**
4:    find $C_i^1$, which have the least assigned feedback;
5:    $m \leftarrow$ the number of assigned feedback in $C_i^1$;
6:    $\mathcal{T}_{fb}^* \leftarrow \mathcal{T}_{fb}$
7:    **for** $j = 1$ **to** $k$ **do**
8:       randomly sample $|C_j^0| \frac{m}{|C_i^1|}$ points in $C_j^0$ to $\mathcal{T}_{fb}^*$;
9:    **end for**
10: **else**
11:    $m \leftarrow |\mathcal{T}_{fb}|$;
12:    $C_i^0 \leftarrow \underset{C \in C^0}{\arg\min} |C|$
13:    **for** $j = 1$ **to** $k$ **do**
14:       randomly sample $|C_j^0| \frac{m}{|C_i^0|}$ points in $C_j^0$ to $\mathcal{T}_{fb}^*$;
15:    **end for**
16: **end if**

---

The main goal of bias-eliminating sampling in the second stage is to judiciously sample from the data selected in the first stage, thereby reducing the distribution bias in the final dataset used for fine-tuning. Our design philosophy regarding the sample size in each cluster is to emulate the distribution of the complete dataset as closely as possible. The number of samples drawn from each cluster is determined by the count of data points in the cluster. If there is no cluster assigned to the feedback data, the minimum number of samples drawn from a cluster equals the total quantity of feedback. The whole process is shown as Algorithm 1.

After sampling, operators are assigned to label each cluster's centroid, determining whether it is an anomaly and extending this judgment to all data within the respective cluster. Although this labeling approach inevitably introduces some errors, labeling each sample individually would be labor-intensive and thus impractical.

## 4 EVALUATION

In this section, we evaluate *AnoTuner* by utilizing the dataset collected from a real-world production environment of a leading ISP. Our goal is to answer the following research questions:

- RQ1: Can *AnoTuner* effectively utilize feedback for fine-tuning and achieve excellent anomaly detection performance at the same time?
- RQ2: How is the quality of the data generated by the false negative augmentation mechanism? Can these data contribute to fine-tuning?
- RQ3: Can the TSAL effectively address the issue of feedback data bias?

## 4.1 Dataset and Evaluation Metric

To verify the effectiveness of our approach, we collected monitoring data from 200 wireless base stations in a certain region provided by a top-tier ISP over a two-week period. The data from the first week serves as the training set, while the data from the second week are used as the test set. Furthermore, we engaged experts in operations and maintenance to label the test set. The dataset includes 25 KPIs covering multiple dimensions of WBS performance (see Appendix A for more details).

The statistic of the dataset mentioned above is shown in Table 3.

Metrics like precision, recall, and the F1-Score are often employed in evaluating time series anomaly detection. Given that operators in practical situations typically disregard point-wise alerts, we adopt an adjustment approach widely used in previous studies [1, 9, 11, 20, 23, 27]. The detail of the adjustment strategy can be found in Appendix C

Since there are multiple WBSes in our dataset, we average the precision and recall across each base station and the F1-Score is calculated based on the average precision and recall.

## 4.2 Setup

In order to effectively evaluate the performance of *AnoTuner*, we chose state-of-the-art (SOTA) MTS anomaly detection models with diverse network architectures as our baselines.

- **ACVAE** [10]: This semi-supervised multi-dimensional time series anomaly detection model is based on two Variational Auto-Encoder (VAE) networks. It features a supervised fine-tuning mechanism, making it the current SOTA model for anomaly detection with a feedback-based fine-tuning mechanism.
- **LSTM-NDT** [4]: This is an unsupervised anomaly detection model based on Long Short-Term Memory (LSTM), widely used in industrial production environments.
- **OmniAnomaly** [23]: Utilizing a Stochastic Recurrent Neural Network as its core, this MTS anomaly detection method exhibits robustness specifically for equipment in data center networks.
- **Interfusion** [11]: This is an unsupervised anomaly detection model based on Hierarchical VAE. It has the capability to simultaneously capture the spatio-temporal relationships of multi-dimensional time series.
- **AnomalyTrans** [28]: This model is based on a Transformer structure and has improved multi-head attention specifically for multi-dimensional time series anomaly detection. It is currently the SOTA model for unsupervised anomaly detection.

It is noteworthy that apart from ACVAE [10], the other anomaly detection baseline models lack meticulously designed fine-tuning mechanisms. Therefore, we can only default to using conventional fine-tuning methods on these models. Moreover, although PUAD [31] is designed with a fine-tuning mechanism, it mainly targets univariate time series anomaly detection and is thus not applicable to anomaly detection in WBSes.

## 4.3 RQ1: Overall Performance

We first evaluate our model's capacity to effectively utilize feedback data for fine-tuning while ensuring excellent anomaly detection performance. Initially, we trained our model preliminarily using the training set within our dataset. As for the test set, we reserve the first 30% in terms of time for feedback data collection, leaving the rest 70% for testing without feedback-based fine-tuning. The main purpose of this partitioning is to simulate real-world WBS anomaly detection scenarios. After employing feedback-based fine-tuning, what operators are most concerned about is the performance in the subsequent online environment. Consequently, our final performance evaluation is conducted using the latter 70% of the test set, while the 30% of the data used for feedback collection is not included in the final performance assessment.

Moreover, to assess the model's capability to learn from feedback data, we divide our experiments into four groups: without feedback tuning (*w/o fine-tuning*), tuning with false positives only (*FP fine-tuning*), tuning with false negatives only (*FN fine-tuning*), and tuning with both FP and FN feedback (*FP+FN fine-tuning*). The final experimental results are as shown in Table 1.

The experimental results indicate that before fine-tuning, the anomaly detection F1-Scores and recalls for *AnoTuner*, ACVAE, AnomalyTrans, and OmniAnomaly can all exceed 0.9. However, after feedback-based fine-tuning, the results change noticeably. AnomalyTrans, OmniAnomaly, and Interfusion all show a decline in the F1-Score after tuning, regardless of whether FP or FN is used for the fine-tuning. The reasons for this situation are twofold. Firstly, these methods lack particular fine-tuning mechanisms and can only use conventional fine-tuning methods for unsupervised anomaly detection, failing to effectively learn from FN data. Secondly, the feedback data are biased and small-scale, leading to model overfitting towards the knowledge contained in the feedback, thereby reducing the robustness of anomaly detection.

When jointly fine-tuning with both FN and FP feedback, the F1-Score of *AnoTuner* increases by 0.0448, while ACVAE, which also possesses a feedback-based fine-tuning mechanism, only increases by 0.0086. The increase in the F1-Score of *AnoTuner* after fine-tuning is about five times that of ACVAE. This affirms that in the context of anomaly detection in WBSes, *AnoTuner* can more effectively utilize feedback data. ACVAE's fine-tuning mechanism mainly learns abnormal patterns through FN data. However, without false negative augmentation, it is difficult for ACVAE to learn insights from the scarce FN feedback. We deduce that the primary reasons why *AnoTuner* can effectively utilize feedback are the resolution of two crucial issues: false negative augmentation mitigates the missing vital information due to the small quantity of FN data, while TSAL tackles the problem of model contamination caused by biased feedback data in fine-tuning.

In summary, compared to SOTA unsupervised anomaly detection models and the semi-supervised model ACVAE, *AnoTuner* utilizes feedback better and is more suitable for conducting anomaly detection in WBSes.

## 4.4 RQ2: Contribution of False Negative Augmentation

The primary goal of the false negative augmentation is to address the issue of a low proportion of false negatives in the already scarce feedback. False negative augmentation can effectively generalize from existing FN data, generating high-quality FN feedback data while ensuring data time and spatial relations.

**Table 1: The precision ($P$), recall ($R$) and F1-Score ($F1$) of *AnoTuner* and baseline methods. *AnoTuner* and ACVAE have a particular mechanism for feedback fine-tuning.**

| Method | w/o fine-tuing | | | FP fine-tuing | | | FN fine-tuing | | | FP+FN fine-tuing | | |
|---|---|---|---|---|---|---|---|---|---|---|---|---|
| | $P$ | $R$ | $F1$ | $P$ | $R$ | $F1$ | $P$ | $R$ | $F1$ | $P$ | $R$ | $F1$ |
| LSTM-NDT | 0.8234 | 0.8926 | 0.8566 | 0.8499 | 0.9074 | 0.8777 | 0.6851 | 0.785 | 0.7317 | 0.8269 | 0.8690 | 0.8474 |
| OmniAnomaly | 0.8964 | 0.9064 | 0.9014 | 0.8718 | 0.8432 | 0.8573 | 0.8529 | 0.7964 | 0.8237 | 0.7293 | 0.7044 | 0.7167 |
| Interfusion | 0.8923 | 0.8786 | 0.8854 | **0.9330** | 0.8321 | 0.8796 | 0.8996 | 0.8164 | 0.8560 | 0.9159 | 0.8187 | 0.8646 |
| AnomalyTrans | **0.9170** | 0.9129 | 0.9149 | 0.8922 | 0.9337 | 0.9125 | 0.8587 | 0.8731 | 0.8658 | 0.8735 | 0.8955 | 0.8841 |
| ACVAE | 0.9158 | 0.8856 | 0.9005 | 0.9022 | 0.8800 | 0.8910 | 0.9230 | 0.8942 | 0.9084 | 0.9222 | 0.8964 | 0.9091 |
| *AnoTuner* | 0.8922 | **0.9411** | **0.9160** | 0.9105 | **0.9510** | **0.9303** | 0.9329 | **0.9707** | **0.9514** | 0.9451 | **0.9770** | **0.9608** |

To evaluate the contribution of false negative augmentation, we first conduct an ablation study. Under the experimental setup of Section 4.3, we compare the results with and without the use of false negative augmentation and compare them with a widely-used temporal augmentation method for time series anomaly detection, RSTL [25]. The experiment results are shown in Figure 4.

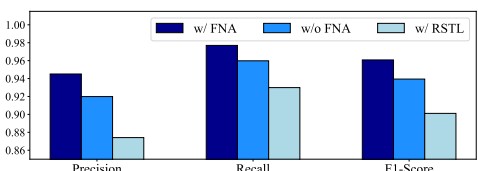

**Figure 4: The performance of *AnoTuner* under three scenarios: with false negative augmentation (w/ FNA), without false negative augmentation (w/o FNA), and using RSTL.**

The results demonstrate that the high-quality data generated by false negative augmentation effectively enhance the effect of feedback-based fine-tuning, leading to an improvement in the model's performance on the final test set. To further analyze the underlying reasons, we conducted an analysis of the data generated by the model. We find that the key to false negative augmentation's success lies in its ability to maintain the temporal and inter-KPI characteristics of the generated data. The anomaly in Figure 5a is that the second curve does not show a similar decline to the first and third KPIs. Figure 5b represents data generated by the false negative augmentation mechanism in *AnoTuner*, while Figure 5c depicts data generated by RSTL. A comparison reveals that the data generated by *AnoTuner* can better ensure the correlation between different KPIs, thus maintaining the physical relationships among various base station KPIs. In contrast, RSTL can generate apparent errors in the interrelationship between the KPIs, distorting the physical meanings of the base station data and thereby preventing the anomaly detection model learning from the data.

We attribute the observed outcome to three main factors. Firstly, RSTL is primarily designed for single-KPI time series and thus lacks the appropriate mechanism to ensure correct interrelationships between KPIs in MTS data augmentation. The second reason, as we see it, is that RSTL lacks a comprehensive understanding of the complete dataset. *AnoTuner*, on the other hand, can view a large amount of WBS data during the training stage, thereby facilitating

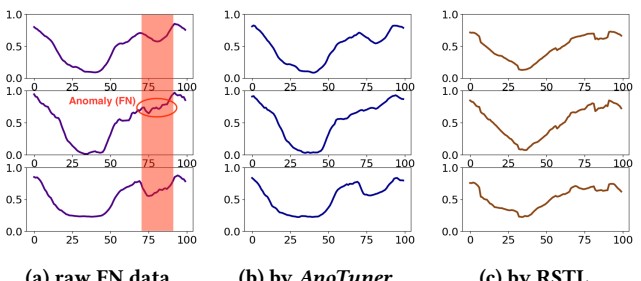

(a) raw FN data      (b) by *AnoTuner*      (c) by RSTL

**Figure 5: The comparison of the generated FN data by *AnoTuner* and RSTL for a FN case (the FN occurs in the red rectangle).**

a better learning of the physical meanings between KPIs. The third point is that these are false negatives (FNs) generated by *AnoTuner* itself, suggesting that *AnoTuner* has inadvertently learned this type of anomalous data pattern during training, thereby generating very high-quality examples.

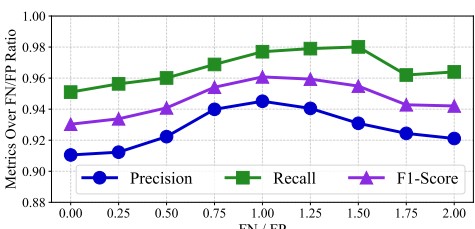

**Figure 6: The performance of *AnoTuner* after fine-tuning at different ratios of FN/FP under the control of false negative augmentation.**

Additionally, we have studied the impact of the ratio of FN to FP on fine-tuning, the results of which are shown in Figure 6. It can be observed from the figure that optimal performance can be achieved when the ratio of FN to FP feedback is approximately 1. Therefore, in our *AnoTuner*, we regulate the false negative augmentation mechanism to maintain the ratio of FN to FP feedback close to 1.

## 4.5 RQ3: Effectiveness of Two-Stage Active Learning

Two-stage active learning (TSAL) is designed to address the problem of degraded anomaly detection performance on online test datasets after fine-tuning due to biased feedback data. In order to evaluate the effectiveness of this mechanism, we conducted an ablation experiment in which we compared the results with and without the mechanism using the experimental setup of Section 4.3, and the results of the experiments are shown in Figure 7.

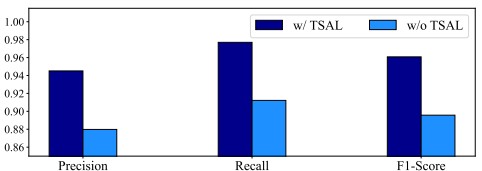

**Figure 7:** *AnoTuner* **performance after feedback-based fine-tuning with (w/) and without (w/o) the use of TSAL**

The result demonstrates that TSAL effectively improves the effect of anomaly detection, which is mainly attributed to the fact that the mechanism effectively solves the problem of biased feedback by sampling from historical data, and makes the fine-tuning process more consistent with the important assumption of the independent identical distribution of machine learning, so the model is not easily contaminated by biased data.

To further analyze how TSAL solves the problem of biased data, we compare the distribution of data before and after active learning and visualize it as shown in Figure 8.

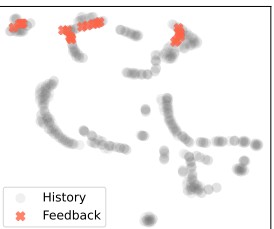 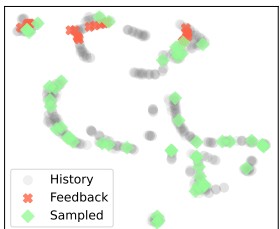

**(a) before sampling**  **(b) after sampling**

**Figure 8: The distribution of the KPI visualized by t-SNE. The grey dots represent the complete dataset, the red ones signify feedback in anomaly detection, and the green ones are obtained through bias-eliminating sampling.**

We randomly sample 500 points from the entire dataset and utilized t-SNE [24] for dimension reduction and conduct 2D visualization. Subsequently, we randomly sample 20 points from the feedback data. As can be observed from Figure 8a, there is a noticeable disparity between the distribution of feedback data points and that of the entire dataset. However, after TSAL, we randomly sample 50 points from the given by the bias-eliminating sampling. The distribution of the dataset used for fine-tuning significantly resembles the distribution of the entire dataset. A distribution closer to that of the entire dataset implies that the model is less likely

to be biased by the feedback data during the fine-tuning process, thereby preventing an overall performance decline.

## 5 RELATED WORK

Anomaly detection in WBSes falls within the domain of multivariate time series anomaly detection. Anomaly detection has been the subject of extensive research over the past decade, particularly in the context of MTS data. Recently, a considerable number of unsupervised deep learning methods have been proposed to target anomaly detection in MTS data, aimed at identifying "data anomalies" within the raw MTS data based on certain assumptions

Generally, these unsupervised methods operate under the premise that "normal patterns" of data originate from a deterministic procedure, and therefore, it is feasible to understand the distribution (or prediction) of these "normal patterns" from raw MTS data. Anomalies are subsequently characterized as data instances that deviate from this learned normal distribution. Various strategies have been employed to achieve this end, including LSTM-based methods like LSTM-NDT [4], multi-head attention-based methods like AnomalyTrans [28], and VAE-based methods such as OmniAnomaly [23], Interfusion [11]. In all these cases, the prediction error or reconstruction error, which signifies the extent of deviation of a data instance from the learned normal patterns, is utilized for anomaly detection.

Unsupervised methods, while useful, often struggle to effectively utilize feedback data in their fine-tuning processes. Recognizing this challenge, recent proposals have introduced semi-supervised methods like PUAD [31] and ACVAE [10]. PUAD combines PU learning techniques and time series clustering to allow the inclusion of labeled data during the fine-tuning process. ACVAE initiates two VAE networks simultaneously, with the anomaly network taking advantage of partially labeled anomaly data. Yet, these methods still have limitations in anomaly detection for WBSes. PUAD is specifically tailored for univariate time series anomalies, which makes it unsuitable for WBS data that is typically composed of MTS. Consequently, PUAD isn't applicable to anomaly detection for WBSes. ACVAE, which primarily learns from data center configurations, also fails to use feedback data effectively. Thus, current anomaly detection methods are still struggling to fully harness the power of feedback data.

## 6 CONCLUSION

In this paper, we proposed *AnoTuner*, a supervised anomaly tuner for unsupervised KPI anomaly detection. Our novel LAELBO loss function and false negative augmentation mechanism enable *AnoTuner* to effectively learn patterns from scarce false negative data. Additionally, the TSAL mechanism proves to be effective to counter the bias introduced by the discrepancy between the distributions of feedback data and that of training data. By sampling a small portion of historical data, the bias could be mitigated, preventing model contamination and thereby significantly enhancing the performance of *AnoTuner*. Experiments on a real-world dataset collected from a top-tier global ISP showcased *AnoTuner*'s performance, even with the feedback data as limited as 0.74% of the test set. This demonstrates the effectiveness of *AnoTuner* in utilizing feedback data to enhance the model's performance.

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

# A  WBS KPI FOR ANOMALY DETECTION

After interviews with ISP operators, we have identified 25 widely used KPIs for WBSes. We categorize these 25 metrics based on their physical meanings. They can be divided into five categories: Business, Physical Resource Block (PRB), Radio Access Bearer (RAB), Handover (HO), and Radio Access Control (RRC) as listed in Appendix A.

**Table 2: An overview of WBS KPIs and their categories.**

| Type | KPI Name | #Number |
|------|----------|---------|
| Business | Paging Congestion Rate, Wireless Connection Rate, Wireless Drop Rate, Handover Success Rate, Upllink Traffic, Downlink Traffic | 6 |
| PRB | Uplink Utilization Rate, Downlink Utilization Rate, Interference Level | 3 |
| RAB | Establishment Success Rate, Establishment Failure Count, Establishment Request Count, Establishment Success Count, Establishment Success Rate, Normal Context Release Count for Requests, Drop Rate, Traffic Volume | 8 |
| HO | Handover Request Count, Handover Success Count, Total Handover Count | 3 |
| RRC | Connection Request Count, Average Connection Count, Maximum Connection Count, Connection Establishment Success Rate, Connection Establishment Success Count | 5 |

# B  DETAIL OF DATASETS

In this paper, we involve three distinct datasets: one sourced from a real-world environment of a top-tier global ISP, while the remaining two are publicly available. For addressing RQ1-3, we sampled 200 WBSes from a top-tier global ISP, collecting the corresponding monitoring data over a two-week period.

For validate the generality of *AnoTuner*, we employ two public multivariate KPI anomaly detection datasets: SWaT (Secure Water Treatment) [16] and WADI (Water Distribution) [8]. These two datasets pertain to water treatment plant operations. Then have been previously utilized for MTS anomaly detection in [8, 11]. Both encompass routine sensor and actuator data from the plants, which form the training set. For the testing set, a combination of normal and anomalous data is present. The anomalies stem from deliberate system attacks introduced to the datasets.

# C  EVALUATION METRIC

Metrics like precision, recall, and the F1-Score are often employed in evaluating time series anomaly detection. Given that operators in practical situations typically disregard point-wise alerts, we adopt an adjustment approach widely used in previous studies [1, 9, 11,

**Table 3: The statistic of three datasets: Wireless Base Station (WBS), Secure Water Treatment (SWaT), and Water Distribution (WADI).**

| Dataset | #Entities | #Metrics | #Train | #Test | Anomaly (%) |
|---------|-----------|----------|--------|-------|-------------|
| WBS | 200 | 25 | 33600 | 33600 | 5.18% |
| SWaT | 1 | 51 | 475200 | 449919 | 12.13% |
| WADI | 1 | 118 | 789371 | 172801 | 5.85% |

**Figure 9: An illustration of the adjustment strategy used in the evaluation metrics. The anomaly points in the ground truth are represented by red rectangles, while the adjusted points are denoted by blue rectangles.**

20, 23, 27]. Given a labeled, continuous anomaly segment, we deem the segment as accurately detected if the method identifies any anomaly within the segment. Consequently, every point within the anomalous segment is regarded as a true positive (TP). If not, every point in the segment is considered a false negative (FN). The points situated outside the abnormal segments are left without adjustment. This strategy is visualized in Figure 9.

# D  EXPERIMENT ENVIRONMENT

All our experiments were conducted on a single-node server equipped with 4 NVIDIA GeForce RTX 3090 GPUs, an Intel(R) Xeon(R) Gold 5218R CPU @ 2.10GHz with 128GB of memory. We implemented our model with Python 3.10.

# E  GENERALITY OF *ANOTUNER*

The multivariate KPI anomaly detection has wide and significant applications in numerous fields [10, 11, 23]. Despite *AnoTuner* being originally conceptualized for WBS scenarios, we postulate its efficacy in diverse domains, contending that with minor adaptations, its applicability extends far beyond its original design. Furthermore, the architectural nuances of *AnoTuner* may potentially serve as a vanguard for innovating anomaly detection methods in related fields. To validate the generality of *AnoTuner*, we introduced two public multivariate KPI anomaly detection datasets, SWaT [16] (Secure Water Treatment) and WADI [8] (Water Distribution). x We adhere to the original division of the training and test sets in the dataset, and retain the settings from RQ1, utilizing the initial 30% of the test set for feedback collection and the latter 70% for performance evaluation. Taking cognizance of the prevalent evaluation standards set by preceding works on these datasets, particularly the frequent adoption of F1-Score as the primary metric [8, 11], we too employ the F1-Score as our metric for evaluation. The pertinent outcomes of our experiments are encapsulated in Table 4.

**Table 4: The F1-Score ($F1$) of *AnoTuner* on datasets from other domains. The first 30% of the test set is used for feedback collection and the latter 70% is used for performance evaluation.**

| Dataset | w/o fine-tuning | FP fine-tuning | FN fine-tuning | FP+FN fine-tuning |
|---------|-----------------|----------------|----------------|-------------------|
| SWaT | 0.8623 | 0.9017 | 0.8874 | 0.9089 |
| WADI | 0.8017 | 0.8345 | 0.8472 | 0.8614 |

Our empirical evaluations underscore the adaptability of our method, reflecting competitive results in anomaly detection across different domains, and even nearing state-of-the-art performances as seen in existing models for the said datasets [11]. An intriguing observation is the relatively subdued improvement due to FN fine-tuning across domains when juxtaposed with its pronounced impact in WBS scenarios. We deduce that the likes of SWaT and WADI exhibit a more balanced sensitivity between FN and FP, which is approximated at a ratio of 1:1. This balance precludes the effective capitalization on False Negative Augmentation. Nevertheless, the feedback-based fine-tuning using TSAL invariably fortifies *AnoTuner* with an augmented capability to assimilate and act upon feedback, leading to an appreciable increase of 5% to 7.5% in F1-Score after fine-tuning. We remain sanguine that the architectural paradigms of our model can be instrumental in advancing anomaly detection methods across various domains.

