# OpenReview forum: "Supervised Fine-Tuning for Unsupervised KPI Anomaly Detection for Mobile Web Systems"
_ACM.org/TheWebConf/2024/Conference — TheWebConf24_

### Official Review · Reviewer_eRDH · 2023-11-21

**Novelty:** 3
**Technical Quality:** 5

**Review:**

In this manuscript, the authors present an approach for Anomaly Detection in Mobile Web Systems. Their approach incorporates a false negative augmentation mechanism to generate similar false negative feedback cases, effectively compensating for the low feedback frequency. In addition, they introduce a Two-Stage Active Learning (TSAL) mechanism that minimizes data contamination issues caused by the difference between the distribution of feedback data and that of the training data. They also disclosed their code anonymously, and the experimental results are convincing. As can be seen from the code file, the model structure is simple and easy to understand. However, there are still many problems with this paper as follows:

* The proposed false-negative enhancement module and TSAL module are not general enough to be used in other anomaly detection methods. This limits the scope of application of the method in this paper.
* The second challenge mentioned in this paper is the problem of data bias. Howeverm, there is a lack of analyzing or demonstrating the severity of its effects. The WBS dataset used for the experiments in this paper only uses 2 weeks of data. In this case, how serious is the "data bias due to hardware and software upgrades and configuration changes" as claimed by the authors?
* This paper uses one real-world dataset and two publicly available datasets. However, there are too few experiments on the two public datasets to demonstrate the generalizability of the methodology.
* There are some typos in the paper, e.g., in Appendix E, there is "x We adhere to the original division of the training and test sets in the 1152 dataset".

**Questions:**

+ Why don't you apply for industry track? I think this article is more suitable for industry track
+ What is the meaning of "Active" in the module named "Two-Stage Active Learning"?

**Ethics Review Description:**

No ethics issue.

**Reviewer Confidence:**

3: The reviewer is confident but not certain that the evaluation is correct

**Scope:**

3: The work is somewhat relevant to the Web and to the track, and is of narrow interest to a sub-community

---

### Official Review · Reviewer_J6wV · 2023-11-22

**Novelty:** 4
**Technical Quality:** 5

**Review:**

This paper presents a system named AnoTuner, which aims to improve anomaly detection performance of the operation status of wireless base stations based on the feedback data.
pros:
1.The authors design a false negative augmentation mechanism, which directly uses the trained AnoTuner to generate more similar feedback based on existing false negative feedback to alleviate the scarcity of the feedback data.
2.The authors design a Two-Stage Active Learning (TSAL) mechanism that samples a small amount from historical data to mitigate the discrepancy between feedback data distribution and training data distribution, effectively avoiding data contamination.
3.The authors have conducted experiments using real-world data collected from a top-tier global ISP. The results show that the performance improvement of AnoTuner after feedback-based fine-tuning is significantly higher than that of the best baseline semi-supervised method.
cons:
1. As the authors highlights that false negative augmentation and TSAL are related to the design of their base model and cannot be directly applied to other anomaly detection methods, the details of the base model are not clarified.
2. Some statements in the paper may make readers confused.
3. The design philosophy of bias-eliminating sampling is not clarified in detail.

**Questions:**

1. The details of the base model in the paper are not clarified. It is better to add the picture of the structure of the base model to the paper, which is shown in the website where the authors' codes are released anonymously.
2. Some statements in the paper may make readers confused. For example, the abbreviation STL may have different meanings. It is better to give the full name "Seasonal-Trend decomposition using Loess" or list the Reference. Besides, in the subsection 2.2, the statement of "Fine-tuning anomaly detection methods using feedback data has been a practice since the era of rule-based anomaly detection" may be confusing. Fine-tuning mainly refers to the technique commonly used in machine learning nowadays, rather than its literal meaning.
3. The procedure of bias-eliminating sampling algorithm is clear, however, the design philosophy of bias-eliminating sampling is not clarified in detail. The reason why the authors choose such a ratio in the algorithm for sampling and how it can emulate the distribution of the complete dataset should be further discussed.

**Reviewer Confidence:**

3: The reviewer is confident but not certain that the evaluation is correct

**Scope:**

3: The work is somewhat relevant to the Web and to the track, and is of narrow interest to a sub-community

---

### Official Review · Reviewer_A5jc · 2023-11-23

**Novelty:** 5
**Technical Quality:** 5

**Review:**

This paper presents AnoTuner, which introduces a false negative augmentation mechanism to generate similar feedback cases about the operation status of wireless base stations (WBSes) and improve anomaly detection performance. A Two-Stage Active Learning (TSAL) mechanism is proposed to minimize data contamination issues arising from distribution differences between feedback and training data. The results show a significantly higher performance improvement.

There are a few weaknesses within this paper. For example, the idea of finetuning with false negative augmented samples is not very novel and clear. There are some settings in experiments not clear enough.

**Questions:**

1. For the novelty of false negative augmentation, only the challenge of MTS data augmentation compared to image augmentation is demonstrated. The challenge of MTS data argumentation in regard to other NLP data augmentation is not clear.
2. Why pick up precision, recall, and F1-score but ignore the accuracy metric?
3. There are lots of similar methods compared with the proposed method. However, why RSTL is used to compare to answer RQ2 but not in RQ1?
4. There are some small errors. For example,
  3.1. Section 4.1 line 588, "multiple dimensions ofWBS performance"
  3.2. Section 4.1 line 590, needs to state that Table 3 is in Appendix.

**Reviewer Confidence:**

2: The reviewer is willing to defend the evaluation, but it is likely that the reviewer did not understand parts of the paper

**Scope:**

4: The work is relevant to the Web and to the track, and is of broad interest to the community

---

### Official Review · Reviewer_ch23 · 2023-11-24

**Novelty:** 5
**Technical Quality:** 6

**Review:**

The authors consider the problem of anomaly detection, motivated by wireless base station monitoring. An issue is that there are very few false negatives, which may be provided from feedback. The main contribution in this paper is to perform fine tuning of an unsupervised classifier based on limited false negative feedback. This is achieved via data augmentation (to increase the number of false alarms) and modifying clustering. With this fine tuning, the method outperforms the state of the art.

The responses of the authors have confirmed my review.

**Questions:**

My question concerns the impact of significant changes to a network configuration, for which the model may not be well-tailored. How does the magnitude of the network configuration impact performance, even with fine tuning? Does the fine tuning have a significant impact in this scenario?

**Reviewer Confidence:**

2: The reviewer is willing to defend the evaluation, but it is likely that the reviewer did not understand parts of the paper

**Scope:**

3: The work is somewhat relevant to the Web and to the track, and is of narrow interest to a sub-community

---

### Decision · Program_Chairs · 2024-01-22

**Decision:**

Accept

**Comment:**

The paper introduces AnoTuner, a system for anomaly detection in wireless base station monitoring. The main contribution is the incorporation of a false negative augmentation mechanism and a Two-Stage Active Learning (TSAL) mechanism. The authors claim significant performance improvement over existing methods through feedback-based fine-tuning. While the experimental results are generally convincing, reviewers bring their concerns about the generality of the proposed mechanisms and potential limitations in addressing data bias.

It's probably worth mentioning that the authors make their code available anonymously, and the model structure is described as simple and easy to understand. The reviewers are leaning towards a positive answer; however, I would like to bring up here the issue around the fit to the scope of track, which was an issue for a similar submission (#2479 Revisiting VAE for Unsupervised Time Series Anomaly Detection: A Frequency Perspective). In this case here, the reviewers actually did not bring this point up, and considered that there is a very good fit with the call for papers. I thus encourage the senior chairs to make an overview pass over the comments from reviewers and the rebuttal phase.